# Extracellular Vesicles Analysis as Possible Signatures of Antiphospholipid Syndrome Clinical Features

**DOI:** 10.3390/ijms26072834

**Published:** 2025-03-21

**Authors:** Giulio Luigi Bonisoli, Giuseppe Argentino, Simonetta Friso, Elisa Tinazzi

**Affiliations:** Department of Medicine, University of Verona, 37134 Verona, Italy

**Keywords:** antiphospholipid syndrome, extracellular vesicles, CD146, CD42a, annexin V, endothelial damage, organ involvement

## Abstract

Antiphospholipid syndrome (APS) is a rare autoimmune disease characterized by thrombosis and obstetric complications. Extracellular vesicles (EVs) of either platelet and endothelial origin are recognized to be involved in the pathophysiology of the disease. This study aimed to evaluate the potential role of endothelial- and platelet-derived extracellular vesicles and the clinical features or progression of APS. We enrolled 22 patients diagnosed with APS and 18 age and sex-matched healthy controls. We determined APS-specific antibody positivity and clinical manifestations in APS affected patients, with a focus on neurological, cardiovascular, dermatological, hematological manifestations, and pregnancy-related complications. Platelet-poor plasma was collected from either patients and controls for the analysis of EVs by flow cytometry technology using monoclonal antibodies to specifically identify those derived from either platelets and/or endothelial cells. EVs of endothelial and platelet origins were overall significantly increased in patients as compared to healthy controls. Furthermore, a significant association was also observed between the number of extracellular vesicles and specific organ involvement, particularly central nervous system manifestations, hematological abnormalities, and obstetric complications. An elevated proportion of endothelial-derived EVs in APS and a reduction of resting endothelial cell-derived EVs were observed in APS-affected women with obstetric complications. Our findings highlight the involvement of endothelial cells and platelets in mirroring the activities of endothelial cells and platelets in APS. Additionally, extracellular vesicles may serve as potential predictors of organ involvement and disease-related damage.

## 1. Introduction

Antiphospholipid syndrome (APS) is an autoimmune disorder characterized by recurrent thrombosis and adverse pregnancy outcomes, accompanied by persistent positivity in antiphospholipid antibody (aPL) assays [1,2]. The updated EULAR classification criteria include several previously considered “non-criteria” manifestations of APS, such as hematological and cardiological features. However, other features, while characteristic, are still excluded but of interest and object of investigation [1,2,3].

aPL represents a heterogeneous group of antibodies that interact with anionic lipids, such as cardiolipin (CL), or with lipid-binding proteins, including tissue factor (TF), annexins, and β_2_-glycoprotein I (β_2_GPI) [2,4,5]. Among all of these antibodies, anti-cardiolipin (aCL) and anti-β_2_GPI (aβ_2_GPI) antibodies are considered pathogenic, likely inducing activation of leukocytes, platelets, endothelial cells, and soluble polymolecular systems [4]. β_2_GPI is strongly associated with APS pathogenesis [4,6], although positivity for aCL and lupus anticoagulant (LA) assays is also related to a higher risk of thrombosis [6]. Additional potentially pathogenic aPLs include anti-annexin V, anti-prothrombin, and anti-phosphatidylserine antibodies [3].

Interactions between aPLs and cellular targets are mediated by cell surface coreceptors such as A5-annexin and A2-annexin [7,8,9], toll-like receptors [7,10,11], and apolipoprotein E receptor 2 (ApoER2) [4,12]. These receptors activate specific cellular functions upon stimulation.

Endothelial cells (ECs) play a central role in APS pathogenesis by entering a state known as “endothelial dysfunction” [13,14]. APS patients exhibit reduced nitric oxide (NO) serum levels and impaired endothelial vascular response [15,16], both referred to as a reduced NO production due to suppression of endothelial nitric oxide synthase (eNOS) activity [13,14]. NO acts as an inhibitor of platelet and leukocyte adhesion and activation; thus, its reduction facilitates thrombosis and inflammation [12,14,16]. Although the exact pathway leading to eNOS inhibition is unclear, evidence suggests involvement of ApoER2 activated by β_2_GPI and aPL [16].

Endothelial dysfunction is also associated with the release of extracellular vesicles (EVs) [13,17,18,19]. These EVs are linked to increased cardiovascular risk and seem to contribute to the pathogenesis of thrombosis and obstetric complications. Dysfunctional endothelium further promotes interactions with immune cells and platelets by upregulating adhesive molecules on endothelial, immune, and platelet surfaces [13]. This dysfunction amplifies immune cell activation, exacerbates systemic inflammation, and increases levels of proinflammatory cytokines such as IL-6 and TNFα. This inflammatory milieu, in turn, up-regulates TF expression [20,21] and release of procoagulant and proinflammatory EVs [22,23].

Circulating EVs, or ectosomes (also referred to as microparticles or microvesicles) [22,24,25], are lipid bilayer-enclosed vesicles ranging from 100 nm to 1000 nm in diameter. These vesicles are released by different cell types, such as platelets, endothelial cells, and leukocytes, through membrane blebbing, which occurs under both physiological and pathological conditions [22,24,25,26]. EVs retain the same membrane as their parent cell, enriched with negatively charged phospholipids [27], and carry cellular components including membrane receptors, proteins, and nucleic acids [22,28,29,30]. Their molecular cargo reflects the characteristics of the origin’s cell, making them identifiable through specific cell surface markers [31]. Functionally, EVs act as intercellular messengers, mediating communication through membrane interactions and transfer of molecules such as proteins and nucleic acids [31,32]. EVs can be divided into two different types on the basis of size, since it is possible to detect “small EVs” characterized by size < 200 nm as well as “medium-large EVs” with size > 200 nm [33,34].

These mechanisms may play a role not only in the primary features of APS but also in its non-criteria manifestations. Although not all clinical features of APS can be directly attributed to a prothrombotic state, some may be linked to underlying autoimmunity or disruptions in intercellular signaling [35].

We decided to focus our attention on small EVs since there are few data about the possible role of medium-large EVs in APS, while the possible pathogenetic role as well as the possible organ involvement predictive value of the small EVs is not yet clear in APS [36,37]. Since we focused our attention only on “small-EVs”, in the text we used the term EVs referring to this specific population of vesicles.

This study aimed to assess differences in EV levels between patients with APS and healthy controls and to explore potential correlations between EV counts and the presence of various clinical manifestations of the syndrome. In addition, we are interested in evaluating whether any correlations between EVs and clinical manifestations may underlie a possible pathogenetic role of the EVs themselves.

## 2. Results

Patients and healthy controls included in the study displayed no differences for clinical characteristics, particularly for age, gender, smoking habit and weight, as well as for considered laboratory data, particularly for CRP, ESR, leucocyte, and platelets count. According to the absence of significant difference between patients and controls, it was not possible to highlight any correlation between EVs and biochemical parameters.

Considering experimental data, in both patients and healthy controls the most abundant population of EVs observed was represented by other platelet-derived EVs (opEVs), followed by resting endothelial cell-derived EVs (reEVs), procoagulant platelet-derived EVs (ppEVs), and apoptotic/activated endothelial cell-derived EVs (aeEVs) (Table 1, Figure 1).

In APS patients, an increased proportion of endothelial-derived EVs (eEVs), including both reEVs and aeEVs, was observed. Specifically, APS patients exhibited a median reEV level of 1.88% (IQR 3.46%) compared to 0.44% (IQR 0.77%) in healthy controls (*p* = 0.01). For aeEVs, the median level in patients was 0.30% (IQR 1.67%) versus 0.00% (IQR 0.00%) in controls (*p* < 0.01).

Platelet-derived EVs (pEVs) also showed distinct patterns of increase in APS patients. The opEVs were markedly elevated, with a median of 22.70% (IQR 16.63%) in patients compared to 12.75% (IQR 9.21%) in healthy controls (*p* < 0.01). ppEVs displayed an increased trend in APS patients, as expected, but did not reach statistical significance.

Considering cardiovascular and hematological features of APS, no significant differences in EV populations were identified, while patients with neurological features exhibited a significant increase in opEVs (*p* = 0.03), with mean levels of 31.61% (±18.37) compared to 19.05% (±8.34) in other patients. A non-significant trend of reduced reEVs was also observed in this group (Table 2).

Women with obstetric manifestations of APS had significantly lower reEV percentages compared to women without pregnancy-related features (*p* = 0.03), with a mean reEV percentage of 0.66% (IQR 0.71), compared to 2.73% (IQR 5.67). Additionally, this subgroup showed a non-significant trend towards reduced ppEV levels compared to other subjects included in the study (Table 3, Figure 2).

Considering the observed trend of EVs in both obstetric and neurological patients, we hypothesized a link between ECD and these clinical features. To explore this hypothesis, patients were grouped based on the presence of clinical features suggestive of ECD. The results are summarized in Table 4, Figure 3. The most notable finding was a slight decrease in reEVs in patients with suspected ECD compared to others (*p* = 0.03).

Overall, patients with hematological, cardiovascular, and obstetric features had higher Disease Complexity Scores (DCS), and those with neurological features exhibited a similar trend, though without statistical significance. No significant differences in EVs populations were assessed across patients with different DCS values. Similarly, no differences in EV levels were observed based on laboratory markers, including antibody positivity or auto-antibodies levels.

Regarding treatment, no significant differences in EV populations were observed between patients receiving warfarin versus acetylsalicylic acid. However, a reduction trend in reEVs was noted in warfarin-treated patients. Other EV populations did not show significant differences in relation to ongoing treatment.

## 3. Discussion

Previous studies have investigated the differences in EV levels between APS patients and healthy controls [36,38,39,40,41], and some data from the literature underlined a possible role for small size EVs as we supposed [33,34,37,42]. However, this study is the first to explore the variations in small EV subsets across different clinical manifestations of APS, including non-criteria features. We have minimized any potentially confusing pre-analytical variable respecting the methodological indications proposed in D MISEV2023. Furthermore, the absence of potential confounding factors for the quantification of EVS, such as smoke, obesity, or inflammation both in the population of patients and the healthy controls represents a guarantee of the adequacy of the data [33,34].

In both patients and healthy controls, the majority of EVs originated from resting cells, while a smaller proportion was derived from activated/dying cells or activated platelets (Annexin V-positive EVs). These findings are consistent with prior research [36,38,39,40,41]. Similarly, in agreement with other studies, we found that eEVs were a minority, while pEVs constituted the largest subgroup of circulating EVs. Our results suggest a narrower gap between EVS levels in APS patients and healthy controls compared to some previous reports [43,44], aligning with the 20% difference described by Arraud et al. [45]. This discrepancy may be attributed to differences in EV isolation methods, which might lack specificity for pEVs or fail to account for EVs derived from megakaryocytes.

The analysis of data obtained in APS patients and healthy controls yielded findings consistent with previous reports [36,38,45]. Our study highlights an increased percentage of eEVs in APS patients. As reported by Dignat-George et al. [36], both aeEVs and reEVs were elevated in patients. These findings suggest that aPL might stimulate endothelial cells, mediating activation or exerting direct/indirect cytotoxic effects. eEVs have been shown to activate different immune cell types, such as plasmacytoid dendritic cells, leading to the production of proinflammatory cytokines [36,38,46]. Additionally, phosphatidylserine-rich EVs can activate granulocytes and enhance cell–cell interactions [46]. These mechanisms could explain ECD, a hallmark of APS, supporting the “first-hit” hypothesis that underlies both thrombotic and non-thrombotic features of the disease [13,14,38].

APS patients also exhibited increased levels of opEVs and ppEVs. The opEVs fraction was nearly doubled in patients, while the ppEVs fraction showed a notable, but statistically non-significant, increase level. Several hypotheses could explain the opEV increase. Firstly, pEVs are known to promote endothelial cell survival and proliferation; in vitro, pEVs enhance endothelial cell survival and the development of capillary-like structures [47,48,49]. Thus, the elevated opEVs in patients may reflect a reactive response to endothelial damage. Secondly, aPL could directly affect platelets by binding receptors such as ApoER2′ and β_2_GPI, leading to a hyperactive state and increased opEV release [4,50]. Thirdly, aPL may destabilize platelet membranes, inducing microparticle release.

The role of ppEVs still remains contentious. These EVs have theoretically a procoagulant role due to phosphatidylserine exposure [51], so higher levels might be expected in APS patients with a history of thrombotic events [52,53]. While some studies have reported differences in ppEV levels between patients with and without thrombosis [52,53], our study displayed no statistically significant differences in ppEV levels between APS patients and healthy controls or between thrombotic and non-thrombotic APS patients. It is possible that the lack of significance is related to the small size of the study population and that the analysis of an increased number of APS patients may lead to more robust as well as different results. On the other hand, since the absence of differences between thrombotic and non-thrombotic patients is reported also by other studies in the literature, it is possible that it may reflect either a lack of sensitivity in ppEV detection methods [27] or the role of ppEVs in thrombus stabilization rather than thrombus initiation. Considering that platelet EVs can enhance thrombin generation [50], a possible alternative explanation of similar levels in patients with and without thrombosis may be explained by this mechanism.

Patients with neurological APS features had significantly higher opEV percentages than those without neurological involvement. Although no prior studies have reported similar findings, we hypothesize that increased platelet activation may contribute to the pathogenesis of neurological events in APS. Additionally, a decreasing trend in reEVs was observed in these patients, and such data may be associated with both arterial-related events and microvascular damage characteristically associated with the syndrome.

The most striking finding was the association between obstetric APS features and reduced reEV levels. Women with obstetric manifestations, such as recurrent miscarriages or infertility, had significantly lower reEV percentages compared to other patients. Even though the small sample size limits definitive conclusions, these findings suggest a potential role for reEVs in placentation and angiogenesis, processes known to be critical during pregnancy and impaired in APS [54]. Reduced reEV levels might represent a “first hit” contributing to conception difficulties and pregnancy loss [55]. Additionally, anti-β_2_GPI antibodies, which are known to impair placentation [13], could further decrease reEV levels.

Considering APS patients based on ECD, the significant reduction in reEV levels reinforces the hypothesis that endothelial damage is central to non-thrombotic APS features. Further studies are needed to confirm these findings, which may highlight the pivotal role of ECD in the disease’s pathogenesis.

No differences in EV populations were observed based on laboratory markers, including antibody type, titer, or isotype. This contrasts with other studies [56], and may be due to the small sample size or a uniform increase across EV subsets, masking compositional differences. Moreover, the absence of relevant differences in clinical features and biochemical data between enrolled patients and healthy controls may also interfere with the homogeneity of data and absence of significance.

Finally, the lack of EV differences in patients with higher Disease Complexity Scores (DCS) raises questions about the utility of this score in differentiating patient subgroups. Similarly, no significant differences were found between patients treated with warfarin and those receiving acetylsalicylic acid, which is consistent with data from the literature [41,55]. These findings underscore the similar effects of these treatments on cellular activity but need further investigation [41,55].

## 4. Materials and Methods

### 4.1. Patients

We enrolled 22 patients diagnosed with APS who were referred to the Autoimmune Diseases Unit, Department of Medicine, University Hospital of Verona. All patients fulfilled the 2023 classification criteria established by the European League Against Rheumatism (EULAR) [55]. The cohort included 18 females and 4 males, with a mean age of 45.5 ± 16 years (range: 20–80). Among the patients, all males and 9 females had primary APS (APS without other autoimmune diseases), while 9 females were diagnosed with secondary APS, occurring in association with other autoimmune conditions. Specifically, 3 had systemic lupus erythematosus, and 4 had undifferentiated connective tissue disease. None of the patients had active thrombosis at the time of enrollment and in particular at the time of blood sampling. The clinical study was conducted by reviewing clinical data from the medical records of patients. According to a recommendation by the International Society for Extracellular Vesicles (ISEV), in addition to positivity of aCL and LA, we also considered leucocytes and platelets count, CRP values, as well as liver and renal function parameters as laboratory data as well as age, gender, smoking habit, or weight as clinical data [34]. We focused on symptoms related to the central nervous system (CNS), cardiovascular (CV) system, dermatological, and hematological manifestations [1,3]. Additionally, we examined pregnancy-related events, including miscarriages, pre-eclampsia, and reduced fertility. Given the absence of a standardized scoring system to assess disease complexity in APS, we developed a simple scoring system, the Disease Complexity Score (DCS), wherein one point was assigned for each organ involvement or damage. Patients were then categorized based on their DCS and affected systems, and we investigated how EV levels varied within these groups.

This study also included 18 healthy controls from laboratory staff and volunteers in the general population, matched to the patient group by age and sex. The control group consisted of 15 females and 3 males, with a mean age of 47 years (range: 30–82). None of the controls had a history or clinical evidence of autoimmune, infectious, cardiovascular, or metabolic disorders. Furthermore, none of the controls were taking any medications at the time of sample collection.

The study protocol adhered to the Helsinki Declaration of 1975 (revised in 2000), and it was approved by the local Ethical Committee (protocol number 1538, version number 3). All patients and controls provided written informed consent to participate to the study protocol.

### 4.2. Flow Cytometric Detection of EVs

Blood samples collected using a Vacutainer system (Becton Dickinson, NJ, USA) were drawn from both patients and healthy controls and were processed within one hour of collection using tubes containing sodium citrate. Platelet-poor plasma (PPP) was obtained following the method previously described by Argentino et al. [19].

EVs in the PPP were analyzed directly by flow cytometry (FACSCanto II, BD Biosciences, San Jose, CA, USA) using monoclonal antibodies [57,58,59]: anti-CD146-PE-CY7 (phycoerythrin-cyanin7) (BD Biosciences, NJ, USA), anti-CD42a-FITC (fluorescein isothiocyanate) (BD Biosciences), and Annexin V-APC (allophycocyanin; AnnV) (BD Biosciences). CD146 was used as a marker for EVs derived from endothelial cells (ECs), as it is expressed on endothelial cells’ surface with higher specificity than CD144 [19,60,61]. CD42a was used to identify platelet-derived EVs [62,63]. Annexin V (AnnV) was chosen as a functional marker, as it binds to phosphatidylserine, which is exposed on the surface of EVs from both apoptotic or activated cells [19,46,64], including activated platelets [65,66]. For each sample, two tubes were prepared: one contained a combination of anti-CD146-PE-CY7 and anti-AnnV-APC, while the other contained anti-CD42a-FITC and anti-AnnV-APC. A morphological gate was established using calibration beads (Flow Cytometry Size Calibration Kit; Life Technologies, Carlsbad, CA, USA) to identify EVs with a diameter of less than 1 μm, allowing their distinction from non-vesicular extracellular particles (NVEPs) as described in the MISEV recommendations [34] (Figure 4). Data analysis was performed using FlowJo software version 10.10 (Tree Star, Ashland, OR USA). The EVs included in the gate were further classified based on the presence of CD146 or CD42a and Annexin V on their surface: CD146-positive EVs were considered endothelial-derived EVs (eEVs), where Annexin V positivity differentiated between apoptotic/activated (aeEVs) and resting endothelial cell-derived (reEVs) groups. CD42a-positive EVs were classified as platelet-derived EVs (pEVs), with Annexin V distinguishing those with procoagulant activity (ppEVs) from others (opEVs).

### 4.3. Statistical Analysis

Statistical analysis was conducted using JASP software version 0.19.3 [67]. Correlations were assessed using a two-tailed Welch’s t-test when the normality assumption was met, accounting for differences in variance, and the Mann–Whitney test was applied in other cases. Statistical significance was defined as *p* ≤ 0.05. Results are presented as mean ± standard deviation for parametric variables and as median with interquartile range for non-parametric variables. All EVs values are expressed as mean percentage of the total population of EVs.

## 5. Conclusions

This study highlights the role of EVs in reflecting the activities of endothelial cells and platelets in APS. These activities appear to be elevated in patients compared to healthy controls, suggesting a role of EVs both of platelets and endothelial origin in cell-cell communication and activation of the disease’s process as previously suggested by Stok and colleagues [37]. In particular, considering the subgroup of APS patients, EV composition varies and seems to correlate with specific disease manifestations. This differential pattern holds potential for predicting complications of the disease, warranting further investigation to validate data.

In women with obstetric features of APS, the significant reduction in reEV levels requires additional study since it may be relevant in the disease’s manifestation development. This finding may reflect a mechanism through which APS disrupts placentation and angiogenesis, contributing to fetal loss and infertility. In case of confirmed data, this insight could enhance understanding of disease pathogenesis and guide future efforts to reduce these complications or modify the therapeutic approach for patients planning pregnancy. Similar considerations apply to neurological events, emphasizing the need for further research in this area. We supposed that inclusion of a greater number of patients could allow a better definition of the relation between EVs and type of neurological damage also in relation to the different laboratory profile of anti-phospholipids tests.

Our data do not support the hypothesis of a different effect in clinical outcome in relation to the presence of alternative therapy with warfarin and acetylsalicylic acid, according to literature. We suppose that the result may be affected by the number of subjects enrolled and the homogeneity of subjects involved, given the absence of significant differences in terms of both clinical profile, including smoking habit and weight, and biochemical characteristics.

A key limitation of this study is the lack of standardization in flow cytometry protocols that complicates comparisons across studies and raises questions about the reliability of results. It is possible that the recent revision proposed by Welsh and MISEV Consortium will surely allow in the future to acquire more homogeneous and comparable data from the different studies. Additionally, the relatively small sample size involved in our study may amplify these concerns, limiting the robustness and generalizability of the findings.

We aimed to evaluate our enhancing statistical power enrolling new subjects affected by APS to improve the number of patients with different clinical phenotypes and validate the observed trends. Furthermore, in vitro and in vivo studies are needed to examine the effects of EVs and aPL on the obstetric and neurological manifestations of APS, providing deeper insights into disease mechanisms and therapeutic opportunities.

## Figures and Tables

**Figure 1 ijms-26-02834-f001:**
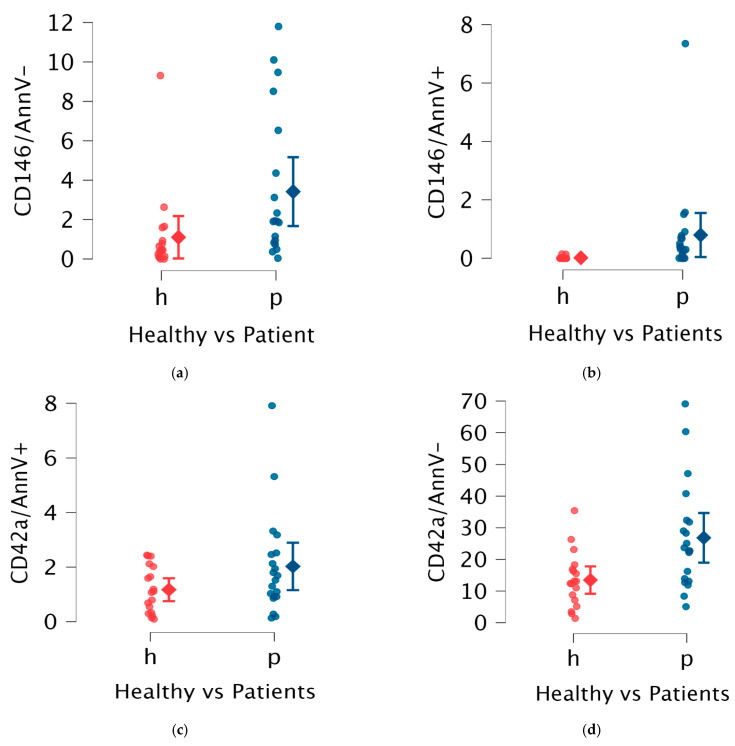
Comparison in EVs between patients (blue marked) and controls (red marked); in particular, in the abscissa axes are included data of healthy controls (h—in red color) and patients (*p*—in blue color) while in the ordinate axes are included surface markers used to define different EVs. (**a**) aeEVs (activated endothelial extracellular vesicles), (**b**) reEVs (resting endothelial extracellular vesicles), (**c**) ppEVs (prothrombotic platelet extracellular vesicles), (**d**) opEVs (other platelet extracellular vesicles).

**Figure 2 ijms-26-02834-f002:**
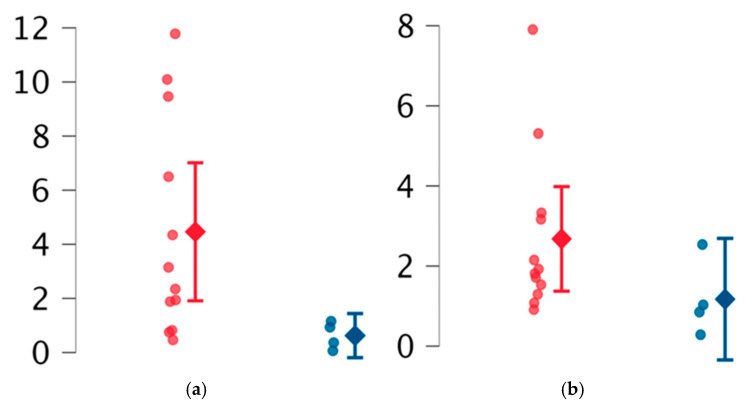
Comparison between women without obstetric features (red marked) and with obstetric features (blue marked). (**a**) reEVs (resting endothelial extracellular vesicles), (**b**) ppEVs (prothrombotic platelet extracellular vesicles).

**Figure 3 ijms-26-02834-f003:**
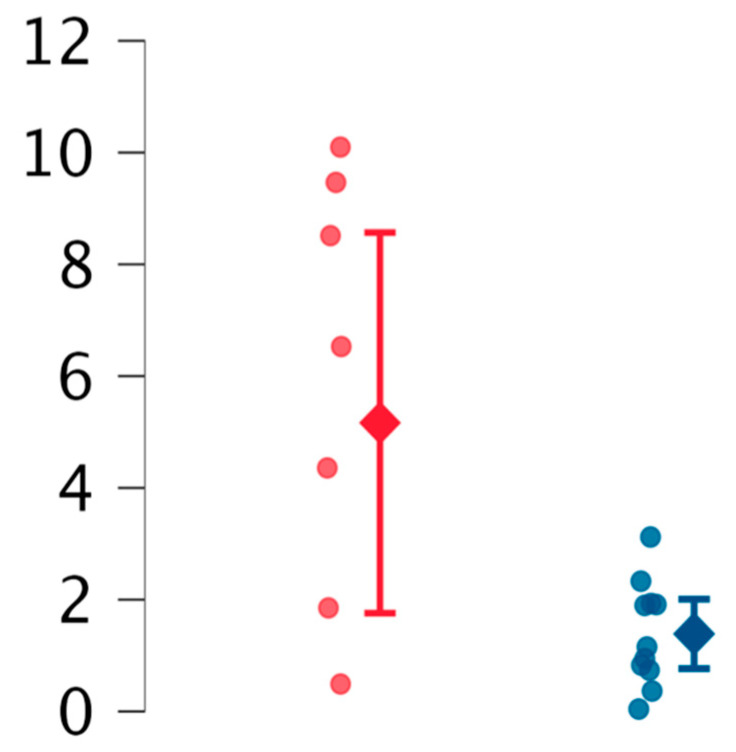
Comparison between reEVs (resting endothelial extracellular vesicles) in patients with suspected ECD (blue marked) and without ECD (red marked).

**Figure 4 ijms-26-02834-f004:**
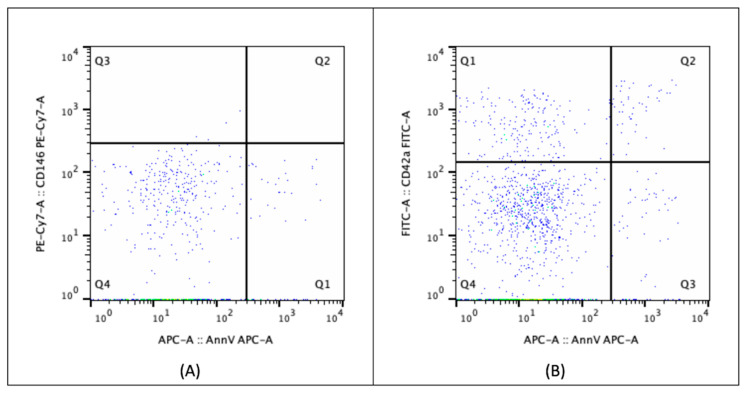
Flow cytometric analysis of extracellular vesicles. Extracellular vesicles (EVs) were selected within a morphological gate corresponding to size range from 100 nm to 1000 nm in diameter. Different subpopulations were identified assessing the expression of CD146 or CD42a and Annexin V on EVs surface: (**A**) CD146-positivity EVs were considered endothelial-derived EVs (eEVs, Q2 and Q3), with Annexin V positivity or negativity differentiating between those derived by apoptotic/activated (aeEVs, Q2) and resting endothelial cell (reEVs, Q3), respectively; (**B**) CD42a-positive EVs were classified as platelet-derived EVs (pEVs, Q2 and Q3), with Annexin V positivity or negativity distinguishing procoagulant ones (ppEVs, Q2) from others (opEVs, Q3).

**Table 1 ijms-26-02834-t001:** Comparison between EVs in patients and healthy controls.

EVs Type	Subgroup	Median EVs	IQR	*p* Value
reEVs	Healthy	0.44	0.77	Mann–Whitney
	Patients	1.88	3.46	0.01
aeEVs	Healthy	0.00	0.00	Mann–Whitney
	Patients	0.30	1.67	<0.01
opEVs	Healthy	12.75	9.21	Mann–Whitney
	Patients	22.70	16.63	<0.01
ppEVs	Healthy	1.09	1.54	Mann–Whitney
	Patients	1.50	1.43	0.156

**Table 2 ijms-26-02834-t002:** Comparison in EVs between patients with neurological features and patients without them. reEVs (resting endothelial extracellular vesicles); aeEVs (activated endothelial extracellular vesicles); opEVs (other platelets extracellular vesicles); ppEVs (procoagulant platelet extracellular vesicles).

EVs Type	Subgroup	Median EVs	IQR	*p* Value
reEVs	Neurological	1.50	1.72	Ns
	Non neurological	3.14	5.72	
aeEVs	Neurological	0.39	0.65	Ns
	Non neurological	0.29	0.48	
opEVs	Neurological	32.77 (mean)	18.68 (SD)	Welch 0.03
	Non neurological	17.89 (mean)	8.01 (SD)	
ppEVs	Neurological	1.61	1.52	Ns
	Non neurological	1.24	2.25	

**Table 3 ijms-26-02834-t003:** Comparison in EVs between women with obstetric features and women without them. reEVs (resting endothelial extracellular vesicles); aeEVs (activated endothelial extracellular vesicles); opEVs (other platelets extracellular vesicles); ppEVs (procoagulant platelet extracellular vesicles).

EVs Type	Subgroup	Median EVs	IQR	*p* Value
reEVs	Obstetric	0.66	0.71	Mann–Whitney
	Non obstetric	2.73	5.67	0.03
aeEVs	Obstetric	0.04	0.23	Ns
	Non obstetric	0.54	0.79	
opEVs	Obstetric	25.50	10.18	Ns
	Non obstetric	27.25	26.75	
ppEVs	Obstetric	0.94	0.69	Ns
	Non obstetric	1.87	1.75	

**Table 4 ijms-26-02834-t004:** Comparison in EVs between patients with clinical features suspected per ECD and without them. reEVs (resting endothelial extracellular vesicles); aeEVs (activated endothelial extracellular vesicles); opEVs (other platelets extracellular vesicles); ppEVs (procoagulant platelet extracellular vesicles).

EVs Type	Subgroup	Median EVs	IQR	*p* Value
reEVs	ECD suspected	1.39 (mean)	0.93 (SD)	Welch
	EC unaffected	5.16 (mean)	4.07 (SD)	0.03
aeEVs	ECD suspected	0.39	0.65	Ns
	EC unaffected	0.30	0.39	
opEVs	ECD suspected	30.49 (mean)	21.51 (SD)	Ns
	EC unaffected	21.16 (mean)	6.48 (SD)	
ppEVs	ECD suspected	1.52	1.35	Ns
	EC unaffected	1.40	1.86	

## Data Availability

The raw data supporting the conclusions of this article will be made available by the authors, without undue reservation.

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
