# Peer review of "Extracellular Vesicles Analysis as Possible Signatures of Antiphospholipid Syndrome Clinical Features"

_ijms, 2025, doi:10.3390/ijms26072834_

Round 1

Reviewer 1 Report

Comments and Suggestions for Authors

Dear Authors,

I've read with great interest your study on Extracellular vesicles analysis as possible signatures of antiphospholipid syndrome clinical features. The study presents valuable insights into the role of extracellular vesicles in antiphospholipid syndrome (APS).

However, I have one major comment, regarding previous studies in this field. It would be beneficial to include a discussion of previously published research on this topic, such as the study by Stok, U., et al., Characterization of Plasma-Derived Small Extracellular Vesicles Indicates Ongoing Endothelial and Platelet Activation in Patients with Thrombotic Antiphospholipid Syndrome. Cells, 2020. 9(5).  This study similarly investigated extracellular vesicles in APS, particularly their endothelial and platelet origins, and reported key findings on their biological properties and implications in endothelial activation and thrombosis. Comparing the present results with those of the earlier study would provide a more comprehensive perspective on the role of extracellular vesicles in APS and strengthen the study’s conclusions.

Minor suggestions

  1.   Methods. Many parameters could confound the EV determination
    and characteristics were recorded at the time of obtaining the blood samples as recommended by the International Society for Extracellular Vesicles (ISEV). Authors should state whether they followed
    Minimal information for studies of extracellular vesicles (MISEV2023): From basic to advanced approaches J Extracell Vesicles,  2024 Feb;13(2):e12404.  doi: 10.1002/jev2.12404.
  2. Figures and tables abbreviations need to be explained i.e. eEVS; apoptotic/activated endothelial cell-derived EVs.
  3. Figure 1,2,3 data should be presented as individual values, not as boxplots, so that the dispersion of data is evident. Also, the individual graphs could be titled for greater clarity.

Author Response

Response to Reviewer 1 Comments

1. Summary

2. Questions for General Evaluation

Reviewer’s Evaluation

Response and Revisions

Does the introduction provide sufficient background and include all relevant references?

Must be improved

We improved the background and also references as suggested. We added more information and details to better define our porpouse for the study.

Are all the cited references relevant to the research?

Yes

Is the research design appropriate?

Yes

Are the methods adequately described?

Must be improved/

We try to better described the methods used for EVs detection, both included MISEV23 suggestions. We used the same methods yet described in our previous study (Argentino et al. as you can see in the bibliography)

Are the results clearly presented?

Must be improved

We added more details in the text

Are the conclusions supported by the results?

Must be improved

We modified our conclusion on the basis of the revision

3. Point-by-point response to Comments and Suggestions for Authors

Comments 1: I have one major comment, regarding previous studies in this field. It would be beneficial to include a discussion of previously published research on this topic, such as the study by Stok, U., et al., Characterization of Plasma-Derived Small Extracellular Vesicles Indicates Ongoing Endothelial and Platelet Activation in Patients with Thrombotic Antiphospholipid Syndrome. Cells, 2020. 9(5).  This study similarly investigated extracellular vesicles in APS, particularly their endothelial and platelet origins, and reported key findings on their biological properties and implications in endothelial activation and thrombosis. Comparing the present results with those of the earlier study would provide a more comprehensive perspective on the role of extracellular vesicles in APS and strengthen the study’s conclusions.

Response 1: Thank you for your suggestions. We have included the study proposed by Stok et al and added some considerations regarding our data. See respectively: Introduction, Line 85-89; Discussion Line 164-168 and 235-240; Conclusions Line 319-325 and 336-341

Comments 2: Methods. Many parameters could confound the EV determination
and characteristics were recorded at the time of obtaining the blood samples as recommended by the International Society for Extracellular Vesicles (ISEV). Authors should state whether they followed Minimal information for studies of extracellular vesicles (MISEV2023): From basic to advanced approaches J Extracell Vesicles,  2024 Feb;13(2):e12404.  doi: 10.1002/jev2.12404.

Response 2: According to you suggestion we have revised part of introduction as well as methods and discussion adding some considerations in respect to the methodological standardization in EVs detection prosposed by the study above reported. See in particular Introduction Line 78-80; Discussion Line 164-172; Materials and methods Line 259 and 263 and Line 294-300; Conclusions Line 342-348.

Comments 3: Figures and tables abbreviations need to be explained i.e. eEVS; apoptotic/activated endothelial cell-derived EVs.

Response 3: Thank you for the suggestions. With the aim to symplified the interpretation of our Tables and figures, we included the explanation of abbreviations in all descriptions included in figures and tables.

Comments 4: Figure 1,2,3 data should be presented as individual values, not as boxplots, so that the dispersion of data is evident. Also, the individual graphs could be titled for greater clarity.

Response 4: As proposed, we modified the figures and now data are presented as individual values instead of boxplots.

Reviewer 2 Report

Comments and Suggestions for Authors

The authors examined several categories of levels in platelet-derived or endothelium-derived extracellular vesicles (EVs) in patients with antiphospholipid syndrome (APS) and investigated their association with patients' central nervous system symptoms, hematologic abnormalities, and obstetric complications. The data and the discussion are well summarized, however, several minor concerns were raised.

  1. The authors collected endothelium-derived or platelet-derived EVs by flow cytometry with CD146 and CD42a antibodies and the size. The presentation of the data of these processes in a figure or a table would be better.
  2. What is the trend of total EVs in patients with APS?
  3. To show the impact of four EVs in patients with APS, the dynamics of the lab data, such as CRP, platelet count, and cytokines, should be included into consideration.
  4. In Table 1-4, ‘reeves’ should be corrected to ‘reEVs’.

Author Response

Response to Reviewer 2 Comments

1. Summary

2. Questions for General Evaluation

Reviewer’s Evaluation

Response and Revisions

Does the introduction provide sufficient background and include all relevant references?

YES

Is the research design appropriate?

YES

Are the methods adequately described?

Can be improved/

We try to better described the methods used for EVs detection, both included MISEV23 suggestions. We used the same methods yet described in our previous study (Argentino et al. as you can see in the bibliography)

Are the results clearly presented?

Can be improved

We added more details in the text

Are the conclusions supported by the results?

YES

3. Point-by-point response to Comments and Suggestions for Authors

Comments 1: The authors collected endothelium-derived or platelet-derived EVs by flow cytometry with CD146 and CD42a antibodies and the size. The presentation of the data of these processes in a figure or a table would be better.

Response 1: Thank you for your suggestions. We have presented the data in new figure included in the study. You can see data presentation in Figure 4.

Comments 2: What is the trend of total EVs in patients with APS?

Response 2: According to you suggestion we have revised part of introduction as well as methods and discussion adding some considerations in respect to the methodological standardization in EVs detection prosposed by the study above reported. See in particular Introduction Line 78-80; Discussion Line 164-172; Materials and methods Line 259 and 263 and Line 294-300; Conclusions Line 342-348.

Comments 3: To show the impact of four EVs in patients with APS, the dynamics of the lab data, such as CRP, platelet count, and cytokines, should be included into consideration

Response 3: Thank you for the suggestions. Data regarding biochemical parameters are now included in different part of the study. In particular you can see modified data in Results Line 96-101; Discussion Line 238-240; Materials and methods Line 256-263; Conclusions Line 319-341

Comments 4: In Table 1-4, ‘reeves’ should be corrected to ‘reEVs

Response 4: Thank you for correction. We modified the text and added the extensive form together the abbreviations in all Table and figures

Round 2

Reviewer 1 Report

Comments and Suggestions for Authors

Dear Authors,

It is evident that you have carefully considered the reviewers' recommendations, and as a result, the article is now significantly improved and much clearer than before. The graphical representations are much more effective and informative, providing a clearer understanding of the content. Additionally, the discussion is now stronger and more engaging, adding further depth to the overall manuscript.

I do have a couple of suggestions for further improvement:

  • Figure 1: The graphs are missing labels for the abscissa and ordinate axes. Adding these labels would enhance clarity.
  • Figure 2: The order of quadrants in Figures A and B seems to differ, which makes the caption below Figure 2a incorrect. I suggest adjusting the caption to reflect the correct order.

Author Response

Point-by-point response to Comments and Suggestions for Authors
Comments 1: Figure 1: The graphs are missing labels for the abscissa and ordinate axes.
Adding these labels would enhance clarity.
Response 1: Thank you for your suggestions. I have modified Figure 1 adding the
description of data in ordinate and abscissa axes. I attach the text below:
Figure 1. Comparison in EVs between patients (blue marked) and controls (red marked);
in particular in abscissa axe are included data of healthy controls (h - in red color) and
patients (p - in blue color) while in ordinate axe is included surface markers used to
define different EVs. (a) aeEVs (activated endothelial extracellular vesicles), (b) reEVs
(resting endothelial extracellular vesicles), (c) ppEVs (prothrombotic platelet extracellular
vesicles), (d) opEVs (other platelet extracellular vesicles)
Comments 2: Figure 2: The order of quadrants in Figures A and B seems to differ, which
makes the caption below Figure 2a incorrect. I suggest adjusting the caption to reflect the
correct order.
Response 2: As proposed, we have modified and correct the description of figure. See the
text below:
Figure 2. Comparison between women patient without obstetric features (red marked) and
with obstetric features (blue marked). (a) reEVs (resting endotelial extracellular vesicles),
(b) ppEVs (prothrombotic platelet extracellular vesicles).
I submit the revised version of the manuscript with corrections in blue color.
